# OpenReview forum: "Image Can Bring Your Memory Back: A Novel Multi-Modal Guided Attack against Image Generation Model Unlearning"
_ICLR.cc/2026/Conference — ICLR 2026 Poster_

### Official Review · Reviewer_yC3h · 2025-10-17

**Soundness:** 1
**Presentation:** 3
**Contribution:** 1
**Rating:** 2
**Confidence:** 4

**Summary:**

The paper introduces RECALL, a reference-guided latent optimization framework designed to expose vulnerabilities in diffusion models after machine unlearning. The method leverages a multi-modal conditioning setup—combining text and image prompts—to adjust a latent representation until the unlearned model regenerates erased concepts. The paper presents the overall motivation, attack pipeline, and evaluation on multiple unlearning strategies (ESD, FMN, UCE, etc.), reporting higher attack success rates than prior works.

**Strengths:**

1. The paper is clearly written and generally easy to follow.

2. The method’s structure and optimization process are well illustrated and explained.

3. The conceptual idea of leveraging latent-space multi-modal guidance for unlearning attacks is mostly novel.

**Weaknesses:**

Mostly concern about the evaluation of whether the method truly works. The current experimental setup fails to convincingly separate genuine recovery of “forgotten” concepts from trivial replay of reference content:

1. The attack initialization already includes 25% of the reference image ($\lambda$=0.25), meaning the optimization starts from a latent that partially encodes the harmful concept itself. This makes the reported ASR potentially inflated and methodologically invalid.

2. The evaluation does not remove trivial copies — if the optimized latent simply reconstructs or memorizes the reference image, ASR no longer reflects a true unlearning breach.

3. The paper does not compare the diversity or distributional coverage of generated samples across methods, leaving it unclear whether RECALL actually recovers a broader concept manifold or merely reproduces a few memorized instances compared to other approaches.

4. The lack of ablations for $\lambda$ approaching 0 (pure noise initialization)  makes it hard to assess whether the method generalizes beyond specific harmful exemplars.


In addition, the paper’s treatment of baselines raises serious concerns. Specifically, the most comparable baseline, UnlearnDiffAtk, has been publicly available since around October 2023, and it is unclear whether any later compatible unlearning attacks were tested. The authors should either include more recent baselines if available, or explicitly clarify in the rebuttal that no newer compatible works exist to justify the current comparison.

Moreover, the authors appear to have **underreported** UnlearnDiffAtk’s performance: its own paper reports 76% ASR (ESD) and 98% (FMN) on Nudity tasks (Table 2), while this paper's Table 1 shows only 51% and 92% respectively. This inconsistency suggests that the baseline reimplementation may be incorrect or incomplete, casting doubt on the claimed relative advantage of RECALL.

**Questions:**

See weakness above.

---

> ### Author Response · Authors · 2025-11-21
> **Response to Official Review by Reviewer yC3h (1/3)**
>
> We sincerely thank the reviewer for the thorough and insightful review. Your comments reflect deep expertise in reinforcement learning and have substantially helped us improve the clarity and rigor of our work. We hope the following responses satisfactorily address your concerns.
>
> ---
>
> >**W1. Noise initialization.**
>
> **A1.** We agree that initializing from a mixture of the reference and noise could, in principle, bias the attack. To check this, we added an ablation over the balancing factor $\lambda$ in Appendix G.2 (Figure 10). When $\lambda=0$ (pure noise initialization), RECALL already achieves strong ASR; increasing $\lambda$ to a small value (e.g., 0.25) further improves ASR only moderately, but substantially reduces the required steps and attack time, with only a minor effect on CLIP alignment. We therefore use $\lambda=0.25$ in the main experiments as a practical trade-off between ASR, semantic alignment, and efficiency, rather than as a way to artificially inflate ASR.
>
> We put part of results (Van Gogh-style, ESD) here:
> | $\lambda$  | 0.00 | 0.10 | 0.15 | 0.20 | 0.25  | 0.30 | 0.35 | 0.40 | 0.45 | 0.50 |
> |------------|------|------|------|------|-------|------|------|------|------|------|
> | ASR        | 76.00| 78.00| 80.00| 84.00| 92.00 | 92.00 | 94.00| 92.00 | 94.00 | 96.00|
> | CLIP Score | 36.00   | 36.20 | 36.10 | 36.40 | 35.92 | 33.60 | 32.10 | 31.40 | 30.80 | 30.20 |
> | Attack Time|124.69 |108.94 | 90.48 | 76.26 | 57.10 | 51.29 | 48.76 | 45.22 | 42.36 | 40.99 |
>
>
> Besides, if simply injecting the reference were enough to make the attack trivially succeed, the Image-only (100% reference) and Text & Image (100% reference) baselines should be the strongest attackers. However, Table 1 shows that both obtain clearly lower ASR than RECALL, and in the diversity analysis (original Table 6 / revised Figure 8 in Appendix F.2, and Table 4 in Appendix D), Image-only often collapses to near-duplicate outputs across tasks and models. In contrast, RECALL uses only a small fraction of the reference at initialization, then optimizes in latent space to find diverse adversarial solutions with higher ASR and lower time cost, which indicates that its effectiveness does not come from trivial copying of the reference.
>
> >**W2. Reconstruct or recall?**
>
> **A2.** RECALL is not designed to replay or reconstruct the reference image; it is used to test whether a target concept that is claimed to be “forgotten” can still be recovered. The reference image only acts as a guidance signal during adversarial image optimization, and all success criteria are defined at the semantic level rather than by similarity to the reference. The reference image itself (or its exact reconstruction) is never counted as an attack sample in our evaluation.
>
> To distinguish RECALL from trivial copying, we provide both semantic and distributional evidence beyond ASR. First, successful samples remain well aligned with the given text prompt (Sec. 5.4), which shows that the generations follow the caption semantics instead of matching the reference appearance. Second, even when using the same reference, the outputs do not collapse to their appearance or layout: the qualitative examples in Appendix D and the DINO-based diversity results in Appendix F.2 show diverse, non-replica generations that still contain the target concept.
>
> Taken together, these observations indicate that RECALL does not simply memorize or reconstruct the reference image. It leverages joint “text+image” conditioning to probe and revive the underlying concept distribution in the unlearned model, so the reported ASR reflects genuine unlearning breaches rather than trivial copies.

---

> ### Author Response · Authors · 2025-11-21
> **Response to Official Review by Reviewer yC3h (2/3)**
>
> >**W3. Add results of diversity across methods.**
>
> **A3.** As clarified in our response to W2, RECALL is designed to revive the underlying concept distribution under a given prompt rather than reproduce a few memorized instances. In the original version, we only compared the diversity of RECALL with Image-only and Text & Image baselines.
>
> In the updated version, we add an explicit cross-method diversity comparison in Appendix F.2 (Figure 8). There, we compute diversity scores for all baselines and RECALL. RECALL consistently shows higher diversity than Image-only and simple multi-modal baselines (Text & Image, Text & R_noise) across unlearned models and tasks, indicating that its outputs do not collapse to copies or trivial transforms of the reference. Compared with strong baselines (P4D (ICML'24)[1], CCE (ICLR'24)[2], UnlearnDiffAtk (ECCV'24)[3], and WACE (NeurIPS'25)[4]), RECALL achieves competitive or higher diversity, with the same trend across different settings.
>
> In addition, qualitative examples in Appendix D and Appendix F.2 further show that RECALL’s recovered images are visually diverse and non-homogeneous while still following the guiding text prompts. Taken together, these results indicate that RECALL recovers a broader concept manifold rather than merely reproducing a few memorized instances.
>
> We put part of the DINO reuslt (RECALL *vs.* baselines) here:
>
> #### DINO – Nudity
>
> |        | Text-only | Image-only | Text & R_noise | Text & Image | CCE  | P4D  | UnlearnDiffAtk | WACE | RECALL |
> |--------|-----------|------------|----------------|--------------|------|------|----------------|------|--------|
> | ESD    | 0.80      | 0.67       | 0.82| 0.69         | 0.57 | 0.62 | 0.69| 0.58 | 0.64   |
> | UCE    | 0.77      | 0.59       | 0.77| 0.72         | 0.69 | 0.62 | 0.66| 0.63 | 0.69   |
>
> ---
>
> #### DINO – Church
>
> |        | Text-only | Image-only | Text & R_noise | Text & Image | CCE  | P4D  | UnlearnDiffAtk | WACE | RECALL |
> |--------|-----------|------------|----------------|--------------|------|------|----------------|------|--------|
> | ESD    | 0.95      | 0.77       | 0.86| 0.85         | 0.89 | 0.90 | 0.90| 0.82 | 0.88   |
> | UCE    | 0.97      | 0.83       | 0.94| 0.94         | 0.90 | 0.88 | 0.91| 0.88 | 0.92   |
>
>
>
> >**W4. Ablation of ($\lambda=0$).**
>
> **A4.** As clarified in our response to W1, in the updated version we extend the ablation of the balancing factor all the way down to $\lambda=0$, which corresponds to pure-noise initialization (**See part of results in Table of W1/A1**, detailed results please refer to Appendix G.2, Fig. 10). The results show that RECALL already achieves strong ASR at $\lambda=0$, and that mixing in a small portion of the reference (e.g., $\lambda=0.25$) yields higher ASR and lower attack time with only a very small change in semantic alignment.
>
> Combined with our clarifications in W2–W3 that RECALL does not memorize or replay specific harmful exemplars, these experiments indicate that RECALL generalizes beyond particular reference images, and that reference-based initialization is an efficiency-motivated design choice rather than a prerequisite for the attack to succeed.

---

> ### Author Response · Authors · 2025-11-24
> **Response to Official Review by Reviewer yC3h (3/3)**
>
> >**Q1.** Add Baseline.
>
> **A1.** Our original strong baselines are P4D (ICML’24)[1], CCE (ICLR’24)[2], and UnlearnDiffAtk (ECCV’24)[3], which remain widely used unlearning attacks in this setting. In particular, UnlearnDiffAtk is still treated as an SOTA attack and is adopted as the default red-teaming tool for new unlearned models in several recent works, including Meta-Unlearning (ICCV’25) [5], MUNBa (ICCV’25) [6], SuMa (ICCV’25) [7], and GLoCE (CVPR’25) [8], where it is explicitly described as state-of-the-art.
> In the updated version, we additionally include the newly published WACE (NeurIPS'25) [9]. With the help from its first author, we reproduced WACE (NeurIPS'25)[4] and reported results for both variants (WACE-N and WACE-C); we put the attack on *Nudity(I2P)* task here, the detalied results of WACE can be found in Sec. 5.2, Sec. 5.3, Sec. 5.4, and Appendix D, E, F.2. Across all these strong and up-to-date baselines, RECALL consistently achieves higher attack success rate, lower attack time, and comparable or higher diversity, which supports the validity of our comparison.
>
> | Method         | ESD  | FMN  | SPM  | AdvUnlearn  | MACE | RECE | DoCo | UCE  | Receler   | ConceptPrune          | Avg. ASR  |
> |----------------|----------------------|----------------------|----------------------|------------------------|----------------------|----------------------|----------------------|----------------------|----------------------|-----------------------|----------------------|
> | P4D-K          | 51.41| 80.28| 76.76| 6.34   | 40.14| 35.92| 77.46| 56.34| 40.14| 77.46| 54.22|
> | P4D-N          | $\underline{62.68}$  | 88.73| 76.76| 2.82   | 32.39| $\underline{52.11}$  | 80.28| 54.93| 35.92| 89.44| 57.61|
> | CCE | 59.15| 85.21| 64.08| $\underline{37.32}$    | $\underline{57.75}$  | 26.76| 30.28| 40.14| 20.42| 83.10| 50.42|
> | UnlearnDiffAtk | 51.41| $\underline{92.25}$  | $\underline{88.03}$  | 8.45   | 47.18| 40.85| $\underline{87.32}$  | $\underline{70.42}$  | $\underline{55.63}$  | $\underline{97.18}$  | $\underline{63.87}$  |
> | WACE-N         | 30.28| 80.99| 61.27| 4.23   | 20.42| 15.49| 58.45| 28.17| 23.24| 80.28| 40.28|
> | WACE-C         | 51.41| 89.44| 79.58| 25.35  | 46.48| 28.87| 71.83| 42.96| 46.48| 88.03| 57.04|
> | **RECALL**     | **71.83** | **100.00**| **96.48** | **60.56**   | **71.83** | **59.86** | **92.25** | **76.76** | **78.87** | **99.30** | **80.77** |
>
>
>
> >**Q2. Results of UnlearnDiffAtk.**
>
> **A2.** We do not underreport UnlearnDiffAtk’s performance. As clarified in our response to Q1, we fully acknowledge that it remains a very strong attack. The fact that some of our numbers (for Nudity) are lower than the original paper, while others (for Church and Parachute) are slightly higher, mainly comes from two factors.
>
> - **Reference images.** We use the official UnlearnDiffAtk code and follow its implementation, but the released repository does not contain the exact reference images used in their paper, so we curate our own reference set. A similar phenomenon also appears for RECALL itself in Appendix F.1 (Table 5): changing the reference image can noticeably affect ASR, so different references naturally yield different numbers.
>
> - **Unlearned checkpoints.** The unlearned models in our experiments come from three sources: AdvUnlearn (NeurIPS'24)[10], weights provided by original authors, and checkpoints we trained by following their official instructions. Minor differences in training details and checkpoint versions can also shift ASR compared with the original UnlearnDiffAtk report.
>
> In summary, the discrepancies are due to differences in references and model weights rather than an incorrect or weakened implementation. All methods, including UnlearnDiffAtk, are evaluated under the same protocol and checkpoints in our paper, so the reported relative advantage of RECALL reflects a fair comparison.
>
> ---
> **Reference**
>
> [1] Prompting4debugging:Red-teamingtext-to-imagediffusionmodelsbyfindingproblematicprompts. In ICML, 2024.
>
> [2] Circumventing concepterasure methods for text-to-imagegenerativemodels. ICLR, 2024.
>
> [3] To generate or not? To Generate or Not? Safety-Driven Unlearned Diffusion Models Are Still Easy To Generate Unsafe Images ... For Now. ECCV, 2024.
>
> [4] WhenAreConceptsErased: When Are Concepts Erased From Diffusion Models? NeurIPS, 2025.
>
> [5] Meta-Unlearning on Diffusion Models: Preventing Relearning Unlearned Concepts. ICCV, 2025.
>
> [6] MUNBa: Machine Unlearning via Nash Bargaining. ICCV, 2025.
>
> [7] SuMa: A Subspace Mapping Approach for Robust and Effective Concept Erasure in Text-to-Image Diffusion Models. ICCV, 2025.
>
> [8] Localized Concept Erasure for Text-to-Image Diffusion Models Using Training-Free Gated Low-Rank Adaptation. CVPR, 2025.
>
> [9] WhenAreConceptsErased: When Are Concepts Erased From Diffusion Models? NeurIPS, 2025.
>
> [10] Defensive unlearning with adversarial training for robust concept erasure in diffusion models. NeurIPS, 2024.

---

### Official Review · Reviewer_iacT · 2025-10-26

**Soundness:** 4
**Presentation:** 4
**Contribution:** 3
**Rating:** 8
**Confidence:** 5

**Summary:**

This paper presented a multi-modal adversarial framework Recall with SOTA results in diffusion model white-box attack settings. The method optimizes the adversarial image in the latent space of the unlearned model itself, requiring no external classifiers. Extensive experiments across ten state-of-the-art unlearning methods and four tasks demonstrate that Recall consistently outperforms existing attacks in success rate, speed, and semantic alignment. The paper shows that current unlearning pipelines are fundamentally fragile against multi-modal adversarial inputs, urging the development of more robust safety measures.

**Strengths:**

1. Recall introduced multi-modal (image+text) attack with the text prompt unmodified, which generates the unlearned image while still keeping semantic fidelity to the original unmodified prompt. The experiment results show SOTA accuracy.
2. Recall is computationally and practically efficient. It doesn't require external models or classifiers. Performing the adversarial optimization directly in the model's latent space is computationally more efficient, which is supported by experiment results.
3. Recall is shows good generalization across models and tasks. It does not overfit to a specific reference image to guide the attack while still producing diverse outputs.
4. Extensive generalization study and ablation study.
5. The paper is well-written, with a clear and compelling narrative from motivation to result.

**Weaknesses:**

1. The paper is more on empirical side. While the results are good, it lacks a theoretical analysis explaining why the multi-modal pathway is so vulnerable or providing formal guarantees about the attack's convergence.
2. The adv_img even though is effective, it will be easily rejected by real image gen system by simple safe guarding before it reaches to the model.
3. adversarial prompt attack was proven to be a good method. what about adversarial prompt + adversarial image, will it get higher ASR? There is no such ablation study in experiment.

**Questions:**

1. 50-step DDIM scheduler is inefficient in general. How will the algorithm work with other faster scheduler?
2. The method focuses on single concept (Van Gogh, or nudity etc.). How about multi-concepts?

---

> ### Author Response · Authors · 2025-11-21
> **Response to Official Review by Reviewer iacT (1/2)**
>
> We sincerely thank you for your highly positive and thoughtful review. Your insightful questions show a deep understanding of our work and have greatly helped us improve both the clarity and rigor of the manuscript. Below, we address each of your comments in detail and provide additional experimental results where appropriate.
>
> ---
>
> >**W1.  Theoretical analysis of convergence guarantees and multi-modal vulnerability.**
>
> **A1.** In this paper, we focus on providing a practical, well-controlled red-teaming tool. To address your concern, the updated Appendix H provides a formal convergence guarantee for the inner-loop latent optimization. For a fixed diffusion step $t$, text embedding $h_t$, and reference latent $z_{\mathrm{ref}}$, the inner loop minimizes
>
> $
> L_{\text{adv}}(z) = \bigl\|F_\theta(z, t, h_t) - F_\theta(z_{\mathrm{ref}}, t, h_t)\bigr\|_2^2,
> $
>
> where $F_\theta$ is the U-Net denoiser. Under standard smoothness and boundedness assumptions on $L_{\text{adv}}$ over a compact latent set, a simplified projected gradient update monotonically decreases $L_{\text{adv}}$ and every accumulation point is a first-order stationary point. Our practical update with momentum and sign-normalization in Eq. (8) is a finite-step approximation of this scheme, and the step-size ablations in Appendix G.1–G.2 are consistent with this behaviour: moderate $\eta \approx 10^{-3}$ yields the best ASR, while much larger or smaller values degrade performance.
>
> For why the multi-modal pathway is vulnerable, we add an explanation rather than a full formal proof. A local linearization of the denoiser around a fixed text embedding suggests a response of the form $W_t h_t + W_i z$, where unlearning mainly weakens the text-driven term $W_t h_t$ for the target concept while leaving large parts of the image-driven term $W_i z$ intact. By fixing $h_t$ and optimizing over the high-dimensional image latent $z$, RECALL searches for directions that reconstruct the forgotten concept and re-align with the text semantics. Results across four tasks, ten unlearning methods, and multiple SD versions (Tables 1–2, 6; Fig. 8) empirically support this picture, and we have clarified in the revised manuscript that a full theory of multi-modal unlearning robustness is left for future work.
>
>
> >**W2. Simple safeguarding.**
>
> **A2.** We agree that online image generation systems may attach front-end safeguards around the model. However, we consider an attacker who has obtained an “unlearned’’ checkpoint and can run it locally (Sec. 3.2, Sec. 4.4). In this case, the attacker will not implement any safeguards to block his desired generation. This is consistent with existing diffusion unlearning evaluations (e.g., P4D (ICML'24)[1], CCE (ICLR'24)[2], UnlearnDiffAtk (ECCV'24)[3], and WACE (NeurIPS'25)[4]). To emphasize, our work focuses on the weaknesses of unlearning models themselves, showing that existing unlearning mechanisms cannot completely unlearn the target concept. This is orthogonal to the external safeguarding solutions.
>
> >**W3. Adversarial prompt + adversarial image.**
>
> **A3.** Text-based attacks that actively modify the prompt typically either change the semantics or require many additional queries, which hurts text–image alignment and significantly reduces efficiency (Sec. 5.4).
>
> Moreover, the benchmarks we use (I2P(CVPR'23)[5], MMA(CVPR'24)[6], ART(NeurIPS'24)[7]) already provide curated adversarial text prompts that are specially designed or optimized to reliably elicit the target concept. RECALL then uses these adversarial prompts as fixed text input and adds an adversarial image condition on top of them, which is conceptually very close to an “adversarial prompt + adversarial image” setting but without an extra text-optimization loop. Under this regime, RECALL already achieves higher ASR, better alignment with the given text, and lower latency than existing text-based baselines, which is particularly attractive for unlearning red-teaming (Sec. 5.2, Sec. 5.3).
>
> We agree that explicitly optimizing both text and image jointly is an interesting extension for an even stronger attacker, but it also raises additional design questions, such as how to keep each modality’s optimization trajectory stable when the other is being updated. A full exploration of such joint optimization is beyond the scope of this paper, and we leave it as future work.

---

> ### Author Response · Authors · 2025-11-21
> **Response to Official Review by Reviewer iacT (2/2)**
>
> >**Q1.** How will the algorithm work with other faster schedulers?
>
> **A1.** We used DDIM-50 mainly to match the commonly adopted community setting and prior unlearning baselines. In the updated version, Appendix G.3 adds a dedicated study where we test three schedulers (DDIM, LMS, PNDM) with 10, 20, 30, 40, and 50 sampling steps (Figure 11). Figure 11(a) shows that, for a fixed number of steps, the three schedulers give almost identical attack success rates (differences are usually less than one percentage point), while increasing the number of steps has a stronger but saturating effect: ASR rises with more steps, but the gain beyond 40 steps is small. Figure 11(b) further shows that most attacks succeed in early steps rather than at the final step of the budget.
>
> Overall, these results indicate that RECALL is essentially scheduler-agnostic and works well with faster schedulers and moderate step counts; using 50 DDIM steps in the main experiments is for fair comparison with existing work, not a requirement of the algorithm.
>
> We put part of the results here:
>
> | Step   | 10     | 20     | 30     | 40     | 50     |
> |--------|--------|--------|--------|--------|--------|
> | DDIM   | 49.94  | 60.56  | 66.16  | 68.46  | 71.83  |
> | LMS    | 49.47  | 61.94  | 64.72  | 69.28  | 71.59  |
> | PNDM   | 48.28  | 60.53  | 67.25  | 69.91  | 72.19  |
>
>
> >**Q2.** Multi-concept attack.
>
> **A2.** We appreciate this question. In line with most existing unlearning and unlearning-attack works, our main experiments focus on single concepts. This is largely because (i) there are very few public checkpoints that reliably support forgetting multiple concepts in one model while preserving overall utility, and (ii) most prior methods are text-based: although one can put several concepts into a single prompt, using multi-label feedback (for example from a multi-label classifier) to jointly optimize the prompt so that multiple targets are recovered without interfering with each other remains technically challenging and underexplored.
>
> For RECALL, the most direct multi-concept extension is to use a reference image (or a small set of references) that contains several target concepts and apply the same multi-modal attack pipeline so that all of them are encouraged to appear in the generated image. Preliminary experiments with a reference containing four targets show this is feasible: we observe per-concept success rates above **60%** for some targets and an overall joint success rate above **30%**. A more systematic study of such multi-concept attacks, and more principled designs beyond this simple extension, is an interesting direction that we plan to pursue in future work.
>
> ---
>
> **Reference:**
>
>
> [1] Prompting4debugging:Red-teamingtext-to-imagediffusionmodelsbyfindingproblematicprompts. In ICML, 2024.
>
> [2] Circumventing concepterasure methods for text-to-imagegenerativemodels. ICLR, 2024.
>
> [3] To generate or not? To Generate or Not? Safety-Driven Unlearned Diffusion Models Are Still Easy To Generate Unsafe Images ... For Now. ECCV, 2024.
>
> [4] WhenAreConceptsErased: When Are Concepts Erased From Diffusion Models? NeurIPS, 2025.
>
> [5] Safelatentdiffusion: Mitigatinginappropriatedegenerationindiffusionmodels. CVPR, 2023.
>
> [6] Mma-diffusion: Multimodalattackondiffusionmodels. CVPR, 2024.
>
> [7] ART: automaticred teamingfortext-to-imagemodelstoprotectbenignusers. NeurIPS, 2023.

---

### Official Review · Reviewer_fUc5 · 2025-10-29

**Soundness:** 4
**Presentation:** 4
**Contribution:** 3
**Rating:** 6
**Confidence:** 3

**Summary:**

This paper introduces RECALL, an unlearning model attack method designed to operate within the image latent space of diffusion models. The core of the method involves using a reference image to guide the iterative generation of an adversarial latent representation ($z_{\text{adv}}$), which successfully recovers a supposedly erased target concept (e.g., specific style or object). The authors conduct extensive experiments to evaluate the method's effectiveness, computational efficiency, and robustness, convincingly revealing significant vulnerabilities in current machine unlearning techniques when subjected to image latent space attacks.

**Strengths:**

1. RECALL successfully identifies adversarial examples in the latent image space, providing compelling evidence that existing unlearning methods (e.g., fine-tuning, knowledge distillation) fail to fully eradicate sensitive or proprietary concepts.
2. The paper includes a comprehensive experimental evaluation and extensive ablation studies that thoroughly assess the potential of image-level attacks on unlearning methods across various metrics and unlearning targets.
3. Efficiency and Practicality: The outstanding experimental results, coupled with a significantly shorter computational time compared to baselines, enhance the practical relevance and real-world applicability of the proposed attack.

**Weaknesses:**

1. Despite the claim of "reference independence" in Section 5.5, the method fundamentally relies on a reference image during the adversarial optimization in Stages I and II. The authors must clarify the specific requirements for this reference image. For instance, what characteristics might a reference image possess that could cause the attack to fail or significantly degrade its performance? Furthermore, given the results in Table 4, which suggest that a simple Image-Only attack can already restore the target concept, the current method appears more like an effective way to refine this recovery by finding a latent state that is minimally destructive to the surrounding concept space, rather than a fundamentally new recovery vector.
2. Insufficient Test Data Coverage: The evaluation is limited by the amount of test data used. To comprehensively assess the robustness of unlearning methods against RECALL, the authors should employ a larger and more diverse dataset.
3. The paper primarily focuses on finding an adversarial latent in the image latent space ($z_{\text{adv}}$) to recover the forgotten concept, with no explicit optimization or guidance related to the textual modality beyond standard conditional inputs. Given this, the claim of presenting a "multi-modal attack" requires further justification or clarification.
4. The claim made in Lines 236-249 seems questionable. RECALL appears to be fundamentally an outcome of a trade-off between prompt following and sampling diversity, which aligns more closely with the diverse sampling results shown in Table 6.
5. The paper would greatly benefit from a brief introductory section or paragraph in the main paper to clarify the common terminology used in this attack space: specifically, the concepts of text-only, image-only, and hybrid/multi-modal attacks/models.

**Questions:**

1. ALGORITHM 1, Line 23 & 24: the final $z_{\text{adv}}$ obtained at Line 23 appears to correspond to the latent state at time $t=0$ (the clean, final image latent). If this is the case, why is $z_{\text{adv}}$ then directly fed back into the diffusion model at Line 24? This procedure deviates from the standard DDPM/DDIM sampling process, which typically starts diffusion from a noisy latent state at $t=T$. Conversely, if $z_{\text{adv}}$ actually corresponds to $t=T$ (a noisy latent), how does the DDIM sampling process manage to produce the diverse recovery results shown in Figure 7?
2. Table 2 CLIP Score Discrepancy: Table 2 shows that the CLIP Score for the images recovered by RECALL is higher than the score for the original Stable Diffusion (SD) model. Please provide an explanation for this phenomenon.
3. The Periodic Integration ablation experiment is incomplete. Specifically, ablation studies are missing for the impact of key hyper-parameters such as the periodic interval and the regularization coefficient.
4. Are LPIPS and IS truly appropriate metrics for evaluating the diversity of the generated images in this attack context? Considering the goal of concept recovery, would a metric based on the variance of DINO scores be a more suitable or complementary choice for measuring recovery diversity?

---

> ### Author Response · Authors · 2025-11-21
> **Response to Official Review by Reviewer fUc5 (1/3)**
>
> We greatly appreciate your positive assessment and thoughtful review. Your questions reflect a strong understanding of this paper and have been instrumental in enhancing both the clarity and rigor of our manuscript.
>
> Below, we have carefully read each of your suggestions and provide detailed responses, accompanied by additional experimental results where appropriate.
>
> ---
>
> >**W1. Clarify the specific requirements for this reference image and the difference between the Image-Only attack and RECALL.**
>
> **A1.** As stated in Sec. 4.2, RECALL assumes a reference image that must contain the target concept (c) and is used to guide the adversarial optimization. In other words, the reference is expected to be semantically aligned with the concept that the attacker aims to generate. If the reference is only weakly aligned or completely unrelated, the attack success rate will naturally degrade because there is insufficient target-specific signal to steer the latent optimization.
>
> In the updated version, we add a dedicated analysis in Appendix G.5. Figure 13 illustrates several new reference choices, and Figure 14 (also in the Table below) reports quantitative results under five conditions: $R_{org}$ (well-matched), Bikini_man and Bikini_woman (partially aligned), and Clothed and Bird (misaligned). The trend is consistent with the above intuition: misaligned references yield inferior outcomes compared to well-matched ones, while partially aligned references sit in between. Unrelated references behave closer to text-only conditioning, leading to lower ASR and higher time cost.
>
> | Ref          | ASR (ESD)           | Step (ESD)             | ASR (UCE)           | Step (UCE)             |
> |--------------|---------------------|------------------------|---------------------|------------------------|
> | ORG          | **71.83**           | **18.65**              | **76.76**           | **16.83**              |
> | Bikini_man   | $\underline{50.96}$ | $\underline{22.03}$    | $\underline{49.25}$ | $\underline{24.56}$    |
> | Bikini_woman | 43.62               | 32.78                  | 35.16               | 30.63                  |
> | Clothed      | 18.62               | 45.86                  | 17.67               | 41.30                  |
> | Bird         | 15.64               | 46.36                  | 15.88               | 40.66                  |
>
>
> Besiedes, regarding Table 4, we emphasize that it is a qualitative visualization and may give the impression that Image-Only already “recovers” the concept in some cases. However, the quantitative results in Table 1 show that Image-Only is in fact a weak attack overall. Unlike Image-Only, RECALL starts from random noise and uses the reference to construct a targeted adversarial image, which is then combined with the original text prompt to query the unlearned model. This staged procedure substantially improves attack success rate, efficiency, and semantic alignment, so RECALL goes beyond merely refining Image-Only and provides a significantly stronger and more reliable tool for probing residual concepts.
>
> >**W2. Broaden Test Data Coverage**
>
> **A2.** We appreciate the suggestion to broaden the test coverage. Our main evaluation is constrained by the availability of publicly released unlearned diffusion checkpoints and their associated tasks, which currently focus on three representative scenarios: nudity, artist style (Van Gogh), and objects (church, parachute) [1,2]. To ensure strict comparability with prior work, we therefore adopt the same datasets and evaluation protocol as UnlearnDiffAtk (ECCV'24) [3].
>
> In the updated version, we introduce two independent and commonly used datasets, MMA(CVPR'24) [4] and ART (NeurIPS'24) [5], under the same protocol (See Appendix C.1). Across both datasets, RECALL consistently achieves higher ASR than strong baselines, indicating that RECALL’s advantage remains robust under larger and more diverse test coverage (*We put Avg.ASR here*, please refer to Sec. 5.2 Table 1 for the detailed results.).
>
> | Avg. ASR | P4D-K | P4D-N | CCE   | UnlearnDiffAtk        | WACE-N | WACE-C | RECALL    |
> |----------|-------|-------|-------|------------------------|--------|--------|-----------|
> | MMA      | 62.70 | 69.51 | 55.70 | $\underline{76.52}$   | 50.23  | 66.25  | **88.20** |
> | ART      | 35.03 | 36.87 | 22.10 | $\underline{43.98}$   | 16.41  | 23.83  | **65.44** |

---

> ### Author Response · Authors · 2025-11-21
> **Response to Official Review by Reviewer fUc5 (2/3)**
>
> >**W3. Clarification of multi-modal attack.**
>
> **A3.** In contrast to mainstream single-modal unlearning attacks that operate purely in the text space (P4D(ICML'24)[6], UnlearnDiffAtk(ECCV'24)[3]), our “multi-modal attack” means that RECALL jointly conditions on both image and text: although the explicit optimization variable is the image latent, every gradient step is taken under the fixed text prompt, so the textual modality continuously participates in noise prediction and constrains the adversarial trajectory.
>
> Under the same evaluation protocol, the text-only and image-only baselines are strictly weaker: take regenerate Church image on unlearned ESD for an example, they either yield lower ASR (16.00%, 4.00%) or produce samples that deviate more from the intended caption (See Table 4 ), whereas RECALL attains higher ASR (*96%*) while preserving the original textual semantics (CLIP Score *27.86*) and maintaining comparable diversity (*0.8842* vs. *0.8023* and *0.6716*). The detailed results can be found in Sec. 5.2, Appendix D, Appendix F.2 in the updated version.
>
> >**W4. Clarification of Lines 236-249.**
>
> **A4.** The sentence in Lines 236–239 was intended to describe the effect of our initialization scheme rather than to claim that RECALL is fundamentally driven by a global prompt–diversity trade-off. The hyperparameter ($\lambda$) mixes a random noise image with the reference image before encoding, which enlarges the region of latent space explored around ($P_{ref}$) and prevents the optimization from collapsing to a single reconstruction of the reference. In practice, we set ($\lambda=0.25$) because it yields stable optimization while avoiding near-duplicate generations around ($P_{ref}$). We clarify this choice and its effect in Appendix G.2 in the updated version.
>
> The quantitative results do not support a simple “more diversity ↔ worse prompt following” trade-off. As shown in Appendix F.2 Figure 8, RECALL achieves clearly higher diversity than the Image-only baseline and compared to other baselines, while Table 2 shows that RECALL also enjoys better semantic alignment with the prompt.
>
> >**W5. Clarify the common terminology.**
>
> **A5.** We appreciate this suggestion. Due to space constraints, our current draft defines the commonly used terms in this attack space in Appendix C.3. In the updated version, we extract and condense these definitions and move them into the main paper, at the beginning of Sec. 5.1 “Baselines”, while keeping the terminology fully consistent with the appendix.
>
> >**Q1. ALGORITHM 1, Line 23 & 24.**
>
> **A1.** The variable obtained at Line 23 is the final adversarial image, which will be used in Stage III (Line 24). In Stage III, we follow the standard multi-modal img2img/TI2I pipeline (provided by the diffusers library) of Stable Diffusion: the model first encodes this adversarial image into a latent ($z_0$), adds noise to reach a starting step ($t_start$), and then runs the usual reverse diffusion process (e.g., DDIM) from this noisy latent, together with the given text prompt, to generate new images.
>
> >**Q2. CLIP Score.**
>
> **A2.** Although the iterative updates in Stages I/II act on the image side, every step is computed under joint text–image conditioning, so the update direction is always shaped by the prompt. As a result, successful attacks naturally yield higher average CLIP similarity for the Nudity and Van Gogh style tasks. By contrast, the original SD baseline performs generic sampling. Even for prompts that explicitly contain nudity-related terms such as "naked" or "nude", only about **95%** of the generated images actually contain nudity (**95.56%** in our measurements). Moreover, Table 2 reports CLIP scores over images from successful attacks that all contain the desired content. So it is expected, rather than contradictory, that RECALL can achieve higher CLIP alignment with the prompt in some cases.

---

> ### Author Response · Authors · 2025-11-21
> **Response to Official Review by Reviewer fUc5 (3/3)**
>
> >**Q3. Complete the ablation studies of periodic intervals and the regularization coefficient.**
>
> **A3.** Thank you for pointing this out. In the updated version, we explicitly ablate periodic integration in Appendix G.4. We vary the *periodic interval* (1, 2, 4, 5, 10, 20) and the regularization coefficient $\gamma$ (from 0 to 0.40). The results in Figure 12 show that moderate intervals between 2 and 5 consistently yield high attack success rates with relatively low sampling cost, while very small (1) or large intervals (10, 20) increase the time cost. For $\gamma$, small-to-moderate values around 0.05 to 0.20 work best; larger values reduce attack success with only minor gains in diversity. Based on these trends, we set the interval to 5 and $\gamma$ to 0.05 in the main experiments.
>
> We put part of results here:
>
> #### Nudity (ESD vs. UCE)
>
> |*periodic*| 1       | 2       | 4       | 5       | 10      | 20      |
> |----------|---------|---------|---------|---------|---------|---------|
> | ESD ASR  | 98.09   | 86.33   | 73.68   | 71.99   | 73.70   | 69.11   |
> | ESD Step | 12.54   | 16.01   | 19.73   | 20.27   | 21.09   | 23.31   |
> | UCE ASR  | 89.23   | 89.53   | 73.89   | 76.83   | 73.70   | 75.24   |
> | UCE Step | 15.90   | 18.55   | 20.50   | 18.54   | 19.55   | 20.13   |
>
> #### Nudity (ESD vs. UCE)
>
> | $\gamma$ | 0.00  | 0.05  | 0.10  | 0.15  | 0.20  | 0.25  | 0.30  | 0.35  | 0.40  |
> |----------|-------|-------|-------|-------|-------|-------|-------|-------|-------|
> | ESD ASR  | 56.77 | 71.99 | 72.41 | 71.65 | 77.86 | 77.86 | 76.27 | 77.78 | 77.78 |
> | ESD DINO | 0.70  | 0.64  | 0.63  | 0.62  | 0.62  | 0.61  | 0.60  | 0.57  | 0.55  |
> | UCE ASR  | 59.83 | 76.76 | 77.18 | 78.31 | 79.98 | 80.32 | 81.41 | 82.64 | 83.18 |
> | UCE DINO | 0.70  | 0.69  | 0.67  | 0.67  | 0.63  | 0.62  | 0.61  | 0.59  | 0.59  |
>
>
>
> >**Q4. Add DINO scores for diversity evaluation.**
>
> **A4.** Thank you for this suggestion. We use LPIPS and Inception Score as our main diversity metrics because they are standard in generative model evaluation [7–10]. We agree that DINO scores provide a complementary semantic view of diversity. In the updated version, we therefore include DINO-based diversity results in Appendix E and Appendix G, compare RECALL with baselines, and ablate key parameters. The trends are consistent with LPIPS and IS: under the guidance of a reference image, RECALL maintains comparable or higher diversity than the baselines while achieving higher attack success rates.
>
> We put part of the DINO reuslt (RECALL *vs.* baselines) here:
>
> #### DINO – Nudity
>
> |        | Text-only | Image-only | Text & R_noise | Text & Image | CCE  | P4D  | UnlearnDiffAtk | WACE | RECALL |
> |--------|-----------|------------|----------------|--------------|------|------|----------------|------|--------|
> | ESD    | 0.80      | 0.67       | 0.82           | 0.69         | 0.57 | 0.62 | 0.69           | 0.58 | 0.64   |
> | UCE    | 0.77      | 0.59       | 0.77           | 0.72         | 0.69 | 0.62 | 0.66           | 0.63 | 0.69   |
>
> ---
>
> #### DINO – Church
>
> |        | Text-only | Image-only | Text & R_noise | Text & Image | CCE  | P4D  | UnlearnDiffAtk | WACE | RECALL |
> |--------|-----------|------------|----------------|--------------|------|------|----------------|------|--------|
> | ESD    | 0.95      | 0.77       | 0.86           | 0.85         | 0.89 | 0.90 | 0.90           | 0.82 | 0.88   |
> | UCE    | 0.97      | 0.83       | 0.94           | 0.94         | 0.90 | 0.88 | 0.91           | 0.88 | 0.92   |
>
>
> ---
>
> **Reference:**
>
> [1] Defensive Unlearning with Adversarial Training for Robust Concept Erasure in Diffusion Models. NeurIPS 2024.
>
> [2] Rethinking Machine Unlearning in Image Generation Models. ACM CCS 2025.
>
> [3] To generate or not? To Generate or Not? Safety-Driven Unlearned Diffusion Models Are Still Easy To Generate Unsafe Images ... For Now. ECCV, 2024.
>
> [4] MMA-Diffusion: MultiModal Attack on Diffusion Models. CVPR, 2024.
>
> [5] ART: Automatic Red-teaming for Text-to-Image Models to Protect Benign Users. NeurIPS, 2024.
>
> [6] Prompt ing4debugging:Red-teamingtext-to-imagediffusionmodelsbyfindingproblematicprompts. In ICML, 2024.
>
> [7] Beyond Aesthetics: Cultural Competence in Text-to-Image Models. NeurIPS, 2024.
>
> [8] CuRe: Cultural Gaps in the Long Tail of Text-to-Image Systems. CVPR, 2025.
>
> [9] Adapting Diffusion Models for Improved Prompt Compliance and Controllable Image Synthesis. NeurIPS, 2024.
>
> [10] Taming Mode Collapse in Score Distillation for Text-to-3D Generation. CVPR, 2024.

---

> > ### Comment · Reviewer_fUc5 · 2025-11-27
> >
> > Thank you for your previous reply. I still have several questions:
> > 1. From Table 1, Text & Image shows the same trend as RECALL (higher attack success → higher RECALL). Does this imply:
> > - (a) Text & Image can expose robustness issues of unlearning methods;
> > - (b) RECALL can be viewed as a Text & Image model with an optimizable image module, leveraging the fact that the image domain contains more difficult-to-erase signals. If so, I do not think it should be called a “multi-modal attack.” Similar to P4D, which attacks mainly in text space but still uses constraints from the latent image space, and is considered a single-modal attack.
> > 2. In Figure 8 The results shows that compared to Text-only, RECALL has lower diversity; compared to Image-only, RECALL has higher diversity. At the CLIP score level, do they follow Image-only > RECALL > Text-only? If so, how do you justify that RECALL does not follow the “higher diversity ↔ worse prompt following” trade-off?
> > 3. Minor suggestions:
> > - (1) In Figure 2, the adversarial image looks like pure noise with no semantic, unlike the examples in Table 7 — this should be adjusted.
> > - (2) In Algorithm 1, line 24 should be expanded (including encoding, noise addition, denoising, etc.).
> > - (3) In Table 1, some datasets (e.g., Nudity-ART) do not show Image-only results.

---

> > > ### Author Response · Authors · 2025-11-27
> > > **Response to follow-up questions**
> > >
> > > Thank you very much for your thoughtful follow-up questions and for giving us the opportunity to further clarify our work. We address your points below.
> > >
> > > ---
> > >
> > > >**Q1. Text & Image and “multi-modal”**
> > >
> > > **A1.** For (a), we agree that the **Text & Image** baseline can also expose robustness issues of unlearning methods. However, its performance remains much weaker than RECALL across all tasks and unlearned models, which motivates going beyond this simple baseline.
> > >
> > > For (b), your understanding is correct: RECALL can be viewed as a **Text & Image** attack setting, but instead of feeding a fixed reference image, we start from a heavily noised image and explicitly optimize it into an adversarial image with the help of the reference. In our terminology, we call RECALL a **multi-modal attack** because it takes both the text and the adversarial image as inputs to induce the unlearned model to regenerate content that is claimed to have been forgotten, achieving high ASR, rather than being a single-modal attack (text-only or image-only). This is different from prior single-modal attacks such as P4D, which only feed an optimized text prompt into the unlearned model.
> > >
> > > ---
> > >
> > > >**Q2. Diversity & CLIP score**
> > >
> > > **A2.** We first clarify the metrics. The diversity in Figure 8 is computed over successful generations that contain the target concept. RECALL indeed has lower diversity than Text-only but higher diversity than Image-only, which shows that using a reference image does not collapse the attack into low diversity.
> > >
> > > For CLIP, we measure the text–image similarity between these successful images and their prompts. Empirically, the ordering is **RECALL > Text-only > Image-only** (there might be small deviations due to the limited number of successful images for Text-only and Image-only). More broadly, diversity and CLIP-based alignment capture different aspects of model behaviour, so higher diversity does not necessarily imply worse prompt following, nor vice versa. Intuitively, Image-only often ignores the prompt (so CLIP is low). Text-only still suffers because unlearning has weakened the link between the target word and the image, so even when the target content can be detected (regarded as successfully generated), the overall semantic alignment is lower than for images generated by the original model. In contrast, RECALL uses the adversarial image to reopen this link and better align the regenerated concept with the given text. Therefore, RECALL achieves higher CLIP alignment without an abnormal loss of diversity.
> > >
> > > ---
> > >
> > > > **Q3. Minor suggestions**
> > >
> > > **A3.** We address minor advices as follows:
> > > - **Figure 2.** In Figure 2, the adversarial image $P_{\text{image}}^{\text{adv}}$ is mainly used to illustrate the overall attack pipeline, while Table 7 shows concrete optimized adversarial images for specific unlearned models and prompts. We have updated Figure 2 to use an example that can successfully induce an unlearned model to regenerate the target content.
> > >
> > > - **Algorithm 1, line 24.** We agree with this suggestion and have expanded Line 24 to explicitly describe the encoding, noise addition, and denoising steps, following the diffusers pipeline more transparently.
> > >
> > > - **Image-only results in Table 1.** For the Nudity unlearning task, although we introduce two additional datasets, the reference image is the same, so we reported the Image-only row only once (*Nudity-I2P*) to avoid redundancy.
> > >
> > >
> > > We hope these clarifications and updates adequately address your remaining questions and help convey the contributions of RECALL more clearly.

---

### Official Review · Reviewer_PMHp · 2025-10-29

**Soundness:** 3
**Presentation:** 3
**Contribution:** 2
**Rating:** 4
**Confidence:** 4

**Summary:**

This paper propose a multi-modal guided attack framework for unlearned diffusion model, where during its attack process only a single reference image is utilized. Also, authors implemented comprehensive experiments on different kinds of adversarial attack and different victim unlearned diffusion model.

**Strengths:**

1. novel multi-modal attack pipeline: latent encoding with reference image blending, iterative latent optimization and the final multi-modal attack using optimized adversarial image with the original text prompt.
2. Strong empirical validation across diverse settings. The evaluation experiments are impressively comprehensive (10 unlearning methods and 3 attack baselines.) The proposed method, RECALL, consistently achieve the best attack performance and also superior semantic alignment.

**Weaknesses:**

1. Authors have overclaimed the independency of their proposed attack method. During attack process, only a single reference image is needed, however, the reference images are still generated by original diffusion models. So, there is an assumption that the original diffusion models are accessible, which cannot be achieved in some cases.
2. Although Appx. F claims the robustness across references, the main text underplays the sensitivity of results to poorly aligned or compositionally distinct reference images. A quantitative failure analysis would clarify generality limits.
3. Some steps resemble prior latent alignment or DreamBooth inversion procedures.

**Questions:**

1. Does RECALL rely on the specific cross-attention fusion mechanism in SD (text-image co-attention), or could it generalize to models like DALLE 3 or Flux that use distinct conditioning pipelines?
2. Could model owners detect such attacks through latent distribution monitoring? If yes, how does RECALL evade simple detection heuristics?

---

> ### Author Response · Authors · 2025-11-21
> **Response to Official Review by Reviewer PMHp (1/2)**
>
> We sincerely thank the reviewer for the thorough and insightful review. Your questions demonstrate deep expertise in this research area and have significantly helped us improve the clarity and rigor of our work.
>
> We hope our explanation can address your concern. Please ask follow-ups if you have any other questions. Thank you!
>
> ---
> >**W1. Reference image relies on the original diffusion model**
>
> **A1.** This is a misunderstanding. RECALL does not require access to any “original model”. The reference image can come from any source such as the public internet/datasets or user-provided assets (see Sec. 4.2); it only needs to contain the target content. In our main experiments, although the reference images were generated, they were produced by third-party diffusion models, not by the original unlearned target model. We further validate source independence by using six other web-sourced references (Appendix F.1, Figure 6); the attack performance remains stable, confirming that RECALL does not depend on any particular source, not to mention the original model.
>
>
> >**W2. Sensitivity to poorly aligned or compositionally distinct references.**
>
> **A2.** Thank you for carefully checking Appendix F.1 and noting the evidence for reference-source independence. We agree that characterizing degradation and failure under “mis-matched” references is important. In the updated version, we add a dedicated analysis in Appendix G.5. Figure 13 presents the newly introduced reference cases, and Figure 14 reports quantitative results across five conditions: $R_{org}$ (matched), Bikini_man, Bikini_woman (partially aligned), Clothed, and Bird (misaligned). The trends are consistent: compared with well-matched references, mis-matched ones yield inferior outcomes due to insufficient target information to guide the attack. Partially aligned references improve the success rate and efficiency relative to misaligned ones, whereas unrelated references behave closer to text-only conditioning, resulting in lower ASR and higher time cost.
>
> | Ref          | ASR (ESD)                | Step (ESD)                 | ASR (UCE)                | Step (UCE)                 |
> |--------------|--------------------------|----------------------------|--------------------------|----------------------------|
> | ORG          | **71.83**                | **18.65**                  | **76.76**                | **16.83**                  |
> | Bikini_man   | $\underline{50.96}$      | $\underline{22.03}$        | $\underline{49.25}$      | $\underline{24.56}$        |
> | Bikini_woman | 43.62                    | 32.78                      | 35.16                    | 30.63                      |
> | Clothed      | 18.62                    | 45.86                      | 17.67                    | 41.30                      |
> | Bird         | 15.64                    | 46.36                      | 15.88                    | 40.66                      |
>
>
>
>
> >**W3. Resemble to prior latent alignment or DreamBooth inversion procedures.**
>
> **A3.** We thank the reviewer for this comment. While RECALL shares some superficial similarities with latent alignment or DreamBooth inversion (e.g., operating in image/latent space and using a reference image), it is fundamentally different in several key aspects.
> - First, RECALL does not fine-tune any model parameters and does not introduce or optimize new text embeddings or special tokens; it operates on a fixed unlearned model and only optimizes an adversarial image latent.
> - Second, the reference image is used solely to provide a stable initialization and guidance signal for regenerating images with target content on the unlearned model, rather than to reconstruct or preserve a specific training example.
> - Third, our goal is not to reconstruct individual memories, but to red-team the unlearned model by probing whether the supposedly forgotten concept can still be generated, how much residual knowledge remains, and how robust the unlearning mechanism is under adaptive attacks.
> - Finally, as shown in Sec. 5.2, Sec. 5.5, and Appendix D and F.2, the images produced by RECALL exhibit substantial diversity, which indicates that our method does not simply recall a single memorized instance but instead explores the remaining concept manifold of the unlearned model.

---

> ### Author Response · Authors · 2025-11-21
> **Response to Official Review by Reviewer PMHp (2/2)**
>
> >**Q1. Generalization to models with distinct conditioning pipelines.**
>
> **A1.** RECALL does not rely on the specific text–image cross-attention implementation in Stable Diffusion, but instead on a higher-level requirement, namely that the generative model accepts joint conditioning on an image and a text prompt. Our method has two key stages: (i) we use a reference image to guide a randomly initialized noisy image toward an adversarial image under the unlearned model, and (ii) we use this adversarial image together with the original text prompt as joint conditions to trigger the unlearned model to produce content that should have been forgotten. Neither step is tied to any particular cross-attention code path; it only requires that the model exposes a pathway where image and text conditions are fused.
>
> We already validate this model-independence within the Stable Diffusion family by showing that RECALL transfers between SD-1.4 and SD-2.x, which supports the claim that the method depends on the existence of a joint conditioning pathway rather than a specific SD cross-attention realization. In principle, Flux models with an img2img pipeline satisfy this requirement, so RECALL can be directly instantiated to evaluate unlearned Flux models. By contrast, the current DALL·E 3 interface only supports text-only API access and does not expose a joint image plus text conditioning interface, so RECALL cannot yet be instantiated on DALL·E 3 in practice now. If a future DALL·E-like model exposes an image plus text conditioning API, the joint-conditioning strategy of RECALL can be ported to that setting as a transfer or query-based attack without modification to the core attack pipeline.
>
> >**Q2. Could model owners detect such attacks via latent-distribution monitoring?**
>
> **A2.** As specified in our threat model (Sec. 3.2), consistent with prior white-box attacks such as P4D (ICML'24)[1], CCE (ICLR'24)[2], UnlearnDiffAtk (ECCV'24)[3], and WACE (NeurIPS' 20'25)[4], we consider an external attacker who has obtained an “unlearned’’ checkpoint and can run it locally. In this case, model owners are not able to collect the runtime latent-distribution statistics from the attacker’s local side.  For the model owner, who uses RECALL as an auditing tool to evaluate the unlearning performance, the goal is to stress-test the raw unlearned model and reveal residual memorization, so there is no need to deploy latent-distribution monitoring in this auditing setting.
>
>
> ---
> **Reference:**
>
> [1] Prompting4debugging:Red-teamingtext-to-imagediffusionmodelsbyfindingproblematicprompts. In ICML, 2024.
>
> [2] Circumventing concepterasure methods for text-to-imagegenerativemodels. ICLR, 2024.
>
> [3] To generate or not? To Generate or Not? Safety-Driven Unlearned Diffusion Models Are Still Easy To Generate Unsafe Images ... For Now. ECCV, 2024.
>
> [4] WhenAreConceptsErased: When Are Concepts Erased From Diffusion Models? NeurIPS, 2025.

---

> > ### Comment · Reviewer_PMHp · 2025-11-26
> >
> > Thanks for the detailed explanations. My concerns have been resolved and I will raise my rating to 6.

---

> > > ### Author Response · Authors · 2025-11-26
> > > **Response to updated rating**
> > >
> > > Thank you very much for your thoughtful follow-up and for taking the time to reconsider your evaluation. We are glad that our additional explanations helped clarify the contributions, and we sincerely appreciate your updated rating.

---

### Author Response · Authors · 2025-11-21
**Common Issue (3/3) – Generation diversity, distributional coverage, and avoiding trivial memorization of the reference.**

> Question (raised by: Reviewer fUc5 – W4; Reviewer yC3h – W2).

> Does RECALL simply reconstruct or memorize the reference image, so that ASR no longer reflects a genuine unlearning breach? Are LPIPS and IS sufficient to evaluate diversity, or should more semantic metrics such as DINO-based diversity be used? Does RECALL actually recover a broader concept manifold compared to other methods?

**Answer 3**
RECALL is **not** designed to reconstruct or replay the reference image. The reference only acts as a **guidance signal** during adversarial optimization, and all success criteria are defined at the **semantic level** (e.g., classifier-based unsafe detection, CLIP alignment with the prompt), not by similarity to the reference. The reference image itself, or its exact reconstruction, is **never counted** as a successful attack sample.

To distinguish RECALL from trivial copying, we provide both **semantic and distributional evidence**. Successful samples remain well aligned with the given text prompt (Sec. 5.4), showing that generations follow the caption semantics rather than the reference’s exact appearance. Even when using the same reference, qualitative examples in Appendix D and the **DINO-based diversity** results in Appendix F.2 show that outputs are visually diverse and non-homogeneous, not near-replicas of the reference.

In terms of metrics, we use **LPIPS and Inception Score** as standard diversity measures, but we also include **DINO-based diversity** in the updated version (Appendix E and G), comparing RECALL to all baselines and ablating key parameters. The trends are consistent: under reference guidance, RECALL maintains comparable or higher diversity than baselines while achieving higher ASR. Cross-method comparisons (Appendix F.2, Fig. 8) show that RECALL has **higher diversity than Image-only and simple multi-modal baselines** (Text & Image, Text & R\_noise), and competitive or higher diversity compared with strong attacks such as P4D (ICML'24)[1], CCE (ICLR'24)[2], UnlearnDiffAtk (ECCV'24)[3], and WACE (NeurIPS'25)[4]. Together with the $λ$-ablation and qualitative results, these findings indicate that RECALL recovers a **broader concept manifold** in the unlearned model rather than reproducing a few memorized instances.

---
**Reference**

[1] Prompting4debugging:Red-teamingtext-to-imagediffusionmodelsbyfindingproblematicprompts. In ICML, 2024.

[2] Circumventing concepterasure methods for text-to-imagegenerativemodels. ICLR, 2024.

[3] To generate or not? To Generate or Not? Safety-Driven Unlearned Diffusion Models Are Still Easy To Generate Unsafe Images ... For Now. ECCV, 2024.

[4] WhenAreConceptsErased: When Are Concepts Erased From Diffusion Models? NeurIPS, 2025.

---

### Author Response · Authors · 2025-11-21
**Common Issue (2/3) – Role and requirements of reference images, and dependence on the original model?**

> Question (raised by: Reviewer PMHp – W1, W2; Reviewer fUc5 – W1).

> Does RECALL fundamentally rely on reference images generated by the original diffusion model? What exactly are the semantic / visual requirements for the reference image? Under what conditions would the attack fail or degrade?

**Answer 2**
RECALL does **not** require access to any original pre-unlearning model. It only assumes a **reference image that explicitly contains the target concept** $c$, and this image can come from any source, such as third-party diffusion models, public datasets, or user-provided assets (Sec. 4.2). In our main experiments, the references are in fact generated by *third-party* models, not by the original unlearned target model, and we further validate **source independence** using six web-sourced references in Appendix F.1, where the attack performance remains stable.

The key requirement is **semantic alignment**: the reference should clearly depict the target concept. In the updated version, Appendix G.5 provides a dedicated analysis with five conditions: R\_org (well-matched), Bikini\_man / Bikini\_woman (partially aligned), and Clothed / Bird (misaligned). The trends are consistent: compared with well-matched references, mis-matched ones yield inferior outcomes due to insufficient target information to guide the attack. Partially aligned references improve the success rate and efficiency relative to misaligned ones, whereas unrelated references behave closer to text-only conditioning, resulting in lower ASR and higher time cost.

---

### Author Response · Authors · 2025-11-21
**Common Issue (1/3) – Clarification of the Threat model: attacker vs. model owner, latent-distribution monitoring and "simple safeguarding".**

> Question (raised by: Reviewer PMHp – Q2; Reviewer iacT – W2).

> Could model owners detect such attacks through latent distribution monitoring? If yes, how does RECALL evade simple detection heuristics? The adv_img even though is effective, it will be easily rejected by real image gen system by simple safe guarding before it reaches to the model.

**Answer 1**
Our threat model follows the standard **white-box unlearning-attack setting** used by (P4D (ICML'24)[1], CCE (ICLR'24)[2], UnlearnDiffAtk (ECCV'24)[3], and WACE (NeurIPS'25)[4]): we primarily consider an **external attacker who has obtained an “unlearned” checkpoint and runs it locally** (Sec. 3.2). In this case, the model owner cannot observe or log the attacker’s runtime latent statistics, so **latent-distribution monitoring is not available as a defense** on the attacker’s side.

When the **model owner** uses RECALL, the role is different: the owner uses RECALL as an **internal auditing / red-teaming tool** (Sec. 3.2, Sec. 4.4) to stress-test whether the unlearning mechanism itself is sufficient. In this auditing regime, the goal is precisely to **expose residual memorization** of the target concept in the unlearned checkpoint; therefore, the owner does not deploy latent-distribution monitoring or simple safeguarding to block their own tests. RECALL is run inside the native multi-modal pathway to reveal remaining weaknesses of the unlearning procedure.

We therefore **do not claim** to evade a particular latent-monitoring or safeguarding defense, nor do we introduce new defenses. Our focus is to show that even under this minimal and widely adopted threat model, existing unlearning mechanisms leave substantial multi-modal vulnerabilities. Designing practical latent-monitoring or front-end filtering strategies against such attacks is an important but **complementary** direction beyond the scope of this attack-focused work.

---
**Reference**

[1] Prompting4debugging:Red-teamingtext-to-imagediffusionmodelsbyfindingproblematicprompts. In ICML, 2024.

[2] Circumventing concepterasure methods for text-to-imagegenerativemodels. ICLR, 2024.

[3] To generate or not? To Generate or Not? Safety-Driven Unlearned Diffusion Models Are Still Easy To Generate Unsafe Images ... For Now. ECCV, 2024.

[4] WhenAreConceptsErased: When Are Concepts Erased From Diffusion Models? NeurIPS, 2025.

---

### Meta-Review · Area_Chair_pMzf · 2025-12-17

**Summary:**

Reviewers raised concerns about both clarity and the strength of the paper’s claims. They questioned whether the proposed attack is truly independent of the original diffusion model and how sensitive it is to the choice and alignment of the reference image, noting that failure cases and degradation under mismatched references were initially underexplained. Several reviewers felt the method resembles prior latent alignment or inversion techniques and asked for clearer differentiation and justification of novelty. Reviewers also questioned whether the approach genuinely recovers a broad erased concept manifold or risks memorizing the reference image, motivating requests for stronger diversity analysis. Additional concerns included the generalizability beyond Stable Diffusion–style conditioning, the realism of the threat model given potential detection or simple safeguards, and the need for more ablations, theoretical insight, and discussion of practical implications for unlearning robustness.

**Reviewer Concerns:**

The rebuttal addressed all major concerns raised by the reviewers. The authors clarified the threat model and the dual role of RECALL as both an external attack and an internal auditing tool, and convincingly addressed questions about dependence on the original diffusion model by adding experiments with third-party and web-sourced reference images. Concerns about sensitivity to reference alignment were largely addressed through new mismatched and partially aligned reference experiments, and the authors strengthened the diversity argument by adding DINO-based diversity metrics, cross-method comparisons, and ablations. The rebuttal further improves differentiation from prior latent alignment and inversion methods. Requests for additional evaluations, baselines, scheduler and step-size ablations, and clearer algorithmic descriptions were also substantially addressed.

Minor concerns remain outstanding, such as questions about real-world practicality—such as detectability under simple safeguards and applicability to models without explicit joint image–text conditioning—are discussed but not empirically resolved. Nevertheless, all responsive reviewers increased their scores to above acceptance threshold.

**Reviewer Scores:**

Reviewer PMHp (score 4)  initially raised concerns about overclaimed independence from the original diffusion model, sensitivity to reference image alignment, and similarities to latent inversion or DreamBooth-style methods. The rebuttal directly addressed these points with clear clarifications, new experiments on reference sources and mismatched references, and a more careful distinction from inversion-based approaches. PMHp explicitly stated that their concerns were resolved and already updated their score from 4 to 6 during the discussion.

Reviewer fUc5 (score 6) was concerned about clarification issues, coverage of evaluation benchmarks, the justification of the “multi-modal” claim, diversity metrics, and several implementation details. The rebuttal responded thoroughly by adding new datasets (MMA, ART), additional baselines, DINO-based diversity analysis, and clearer explanations of Algorithm 1 and design choices. Given that fUc5 did not raise further objections after the follow-up and most concerns were addressed, this reviewer would likely maintain their score at 6, or possibly increase slightly, but probably not as much as to upgrade the score to 8.

Reviewer iacT (score 8) requested deeper theoretical justification, more ablations over schedulers and optimization budgets, analysis of joint text–image optimization, and discussion of broader applicability and safeguards. The rebuttal added convergence discussion, extensive scheduler and step ablations, and expanded discussion on extensions and threat modeling. While the added material addresses most requests, the theoretical component remains relatively light. As a result, iacT would likely keep the score.

Reviewer yC3h (score 2) expressed the strongest concerns, questioning whether RECALL merely replays or memorizes the reference image, whether diversity metrics were sufficient, and whether baselines such as UnlearnDiffAtk were fairly represented. Although the rebuttal added new diversity analyses (including DINO-based metrics), noise initialization ablations, and an additional strong baseline (WACE), the core concern about definitively ruling out reference memorization (W4) is only partially resolved. Given this, yC3h would likely raise the score modestly, but still remain skeptical—most plausibly increasing from 2 to 3 or 4, rather than fully aligning with the other reviewers’ more positive assessments.

The AC is convinced by the majority of positive reviewer feedback and the substantial improvements made during the rebuttal and discussion period. While Reviewer yC3h’s W4 concern regarding potential reference memorization was only partially resolved, the AC does not view this issue as a critical blocker that would justify rejection, especially in light of the added diversity analyses, ablations, and strengthened evaluation overall. The AC strongly encourages the authors to fully incorporate all new experimental results, analyses, and clarifications introduced during the rebuttal into the camera-ready version.

---

### Decision · Program_Chairs · 2026-01-26

Accept (Poster)